Gamify4LexAmb: a gamification-based approach to address lexical ambiguity in natural language requirements

Dar Hafsa 1
Aziz Romana 2
Khan Javed Ali 3
Lali Muhammad IkramUllah 4
Almujally Nouf Abdullah naalmujally@pnu.edu.sa 2
1 Department of Software Engineering, University of Gujrat , Gujrat , Punjab , Pakistan
2 Department of Information Systems, College of Computer and Information Sciences, Princess Nourah bint Abdulrahman University , Riyadh , Saudi Arabia
3 Department of Computer Science, School of Physics, Engineering and Computer Science, University of Hertfordshire , Hertfordshire , United Kingdom
4 Department of Information Sciences, University of Education , Lahore , Punjab , Pakistan
Sohaib Osama
Electronic publication date: 2024 Sep 19
Publication date: 2024
Volume: 10
Electronic Location ID: e2229
Received 2024 May 3; Accepted 2024 Jul 11
Copyright: ©2024 Dar et al.
Copyright year: 2024
Copyright holder: Dar et al.
License: This is an open access article distributed under the terms of the Creative Commons Attribution License, which permits unrestricted use, distribution, reproduction and adaptation in any medium and for any purpose provided that it is properly attributed. For attribution, the original author(s), title, publication source (PeerJ Computer Science) and either DOI or URL of the article must be cited.
License URL: https://creativecommons.org/licenses/by/4.0/

Keywords: Requirements elicitation, Gamification, Natural language, Requirements ambiguity, Lexical ambigutiy, Game elements, POS tags, PBL, Ambiguity detection, User engagement

Funding: Princess Nourah bint Abdulrahman University Researchers Supporting Project number PNURSP2024R410 Princess Nourah bint Abdulrahman University, Riyadh, Saudi Arabia This work was supported by Princess Nourah bint Abdulrahman University Researchers Supporting Project number (PNURSP2024R410), Princess Nourah bint Abdulrahman University, Riyadh, Saudi Arabia. The funders had no role in study design, data collection and analysis, decision to publish, or preparation of the manuscript.

==============================
Ambiguity is a common challenge in specifying natural language (NL) requirements. One of the reasons for the occurrence of ambiguity in software requirements is the lack of user involvement in requirements elicitation and inspection phases. Even if they get involved, it is hard for them to understand the context of the system, and ultimately unable to provide requirements correctly due to a lack of interest. Previously, the researchers have worked on ambiguity avoidance, detection, and removal techniques in requirements. Still, less work is reported in the literature to actively engage users in the system to reduce ambiguity at the early stages of requirements engineering. Traditionally, ambiguity is addressed during inspection when requirements are initially specified in the SRS document. Resolving or removing ambiguity during the inspection is time-consuming, costly, and laborious. Also, traditional elicitation techniques have limitations like lack of user involvement, inactive user participation, biases, incomplete requirements, etc. Therefore, in this study, we have designed a framework, Gamification for Lexical Ambiguity (Gamify4LexAmb), for detecting and reducing ambiguity using gamification. Gamify4LexAmb engages users and identifies lexical ambiguity in requirements, which occurs in polysemy words where a single word can have several different meanings. We have also validated Gamify4LexAmb by developing an initial prototype. The results show that Gamify4LexAmb successfully identifies lexical ambiguities in given requirements by engaging users in requirements elicitation. In the next part of our research, an industrial case study will be performed to understand the effects of gamification on real-time data for detecting and reducing NL ambiguity.

Introduction

Natural language requirements are prone to inherited complexities such as vagueness, incompleteness, and ambiguity. According to previous studies, requirements ambiguity is a more serious problem as compared to other problems of requirements (Alvertis et al., 2016). Requirements ambiguity is referred to as ‘uncertainty’ or ‘misinterpretation’ of the context, whereas IEEE quotes a practice for software requirements specifications (SRS) that “An SRS in unambiguous if and only if every requirement stated therein has only one interpretation” (Bano, 2015). Not only is ambiguity itself challenging in requirements, but it also results in incomplete and inconsistent requirement specifications (Kamsties, 2005). These inconsistencies are passed on to the other stages of software development (Luisa, Mariangela & Pierluigi, 2004), resulting in faulty and erroneous software systems.

In literature, ambiguity is classified into different types and categories. Researchers have introduced several taxonomies for classifying ambiguity in natural language, including taxonomy by Bano (2015), which is notable and important because it provides a comprehensive classification of natural language ambiguity covering different types of ambiguity, including lexical, semantic, syntactic, and pragmatic (Berry, Kamsties & Krieger, 2003). This contextual categorization caters to improving natural language requirements if the process is in place for its detection and resolution early in the software development activities, i.e., requirements analysis and specification. Each type of ambiguity is further divided into sub-types, as shown in Fig. 1.

Figure 1 Classification of ambiguity types.

As shown in Fig. 1, each type of ambiguity addresses different problems of natural language requirement. Among other types of ambiguity, such as semantic, syntactic, and pragmatic, lexical ambiguity is significant due to its significant impact on language understanding. For non-specialists, lexical ambiguity is a language barrier (Liu, Medlar & Głowacka, 2022); moreover, for empirical evaluations, lexical ambiguity is not as prominent as syntactic and semantic ambiguities are (Bano, 2015). However, the scope of our study is limited to lexical ambiguity. For the proposed approach, we emphasize the identification of lexical ambiguity in the end-user requirements. Lexical ambiguity is significant in natural language requirements due to its impact on requirements’ precision and clarity. In lexical ambiguity, a word has more than one meaning, which leads to misunderstandings and confusing statements, resulting in challenges for software developers when implementing the requirements (Berry, 2008). We argue that addressing lexical ambiguity can improve the accuracy of requirements, reduce the risk of project failure and facilitate communication among various stakeholders.

Lexical ambiguity

Lexical ambiguity occurs when a word possesses several meanings (Berry & Kamsties, 2004). For instance, in the example ‘Develop a bank interface for checking account balances’, the term ‘interface’ is ambiguous because it has multiple interpretations (such as user interface or technical interface). Therefore, the ambiguity in the term ‘interface’ may lead to misunderstood requirements. Semantic ambiguity is further divided into homonymy and polysemy ambiguities. Homonymy occurs when two different words have unrelated meanings and etymology (history of development) but the same written and phonetic representation (Beg, Abbas & Joshi, 2008). For instance, ‘bank’ in the context of custody (issue of money, loan, exchange) or ‘bank’ in the context of rising ground bordering (lake, river). Polysemy occurs when a word has several related meanings but one etymology (Berry, 2008). For instance, ‘green’ is green; youthful; not ripened. According to literature (Shan & Mutty, 2022), some work has been proposed on lexical expressions in requirements to lead other classes of ambiguity, but overall lexical ambiguity has not been focused much in the requirements engineering (RE) community (Gleich, Creighton & Kof, 2010), according to our knowledge. Where natural language (NL) requirements tend to be ambiguous, and for this reason, the SRS document requires pre-processing with natural language processing (NLP) techniques for the detection of ambiguity in requirements (Berry, Kamsties & Krieger, 2003; Berry, 2008). However, manual ambiguity resolution of requirements is time-consuming, error-prone, and costly (Bano, 2015). Moreover, it does not involve users at the time of requirements elicitation.

Gamification

Recently, researchers have been working on more inclusive techniques to involve stakeholders in the process of RE. Among such techniques is gamification which helps users to get engaged during elicitation and provide their requirements (Alsawaier, 2018). Gamification enhances user engagement by introducing them to fun ways in non-fun environments (Burke, 2012). Results of previous studies have shown that the participants involved in gamified systems agree with the significance of gamification, such as overcoming passiveness, boredom, and repetition. A major part of gamification is the use of interesting game elements that motivate users to perform the desired task without losing interest in the activity. Commonly used game elements are points, badges, leaderboards (PBL) (Deterding et al., 2011), levels, avatars (Dar, Imtiaz & Ullah Lali, 2022), stories (Tondello, Mora & Nacke, 2017), and others. The game elements are selected in the design phase of the system based on user roles and the functionality of the system.

Research motivation

In the literature, there are several methods proposed that identify and remove ambiguity in requirements (Bano, 2015), but limited research is reported that involves users during elicitation (Sinha & Husain, 2016) for reducing ambiguity in requirements (Preston, 2014), according to our knowledge, to date. The existing approaches lack in encouraging user participation and have non-friendly and non-interactive interfaces. In contrast, user engagement in the elicitation not only helps to elicit requirements but also reduces ambiguities by actively involving users in the initial requirements engineering phase (Dar, 2020). The motivation behind this work is to propose a gamified elicitation tool that engages users during requirements elicitation to detect and reduce requirements ambiguity. In this way, gamification can enhance user engagement, and users become helpful to development teams in reducing ambiguity during elicitation, reducing cost and time. For this purpose, we have designed game elements and rules for the proposed system, along with the ambiguity rules (Deterding et al., 2011; Dar, 2020), to identify lexical ambiguity. In this work, we focus on raw NL requirements collected during elicitation that still need to be documented in SRS. The contributions of this work are:

i. Designing a framework for handling lexical ambiguity in NL requirements

ii. Lexical ambiguity detection using ambiguity rules and a dictionary

iii. Lexical ambiguity reduction (by engaging users in requirements elicitation using gamification)

iv. Performed by domain experts from industry

v. Developed an initial prototype

Furthermore, we formulated the following research questions to systematically achieve the aim of this study.

RQ-1: How can lexical ambiguities be eliminated from NL requirements while eliciting end-user requirements?

RQ-2: How well does the Gamification for Lexical Ambiguity (Gamify4LexAmb) prototype align with the theoretical foundations of this study?

For answering RQ-1, we designed a framework based on ambiguity rules, parts of speech (POS) tags, and word dictionaries. Firstly, POS tags identify proper nouns in each NL requirement. The domain-specific dictionary of words and man-made ambiguity rules are used. The rules are embedded in a Gamify4LexAmb framework. For RQ-2, the Gamify4LexAmb prototype is developed to validate the Gamify4LexAmb framework. We evaluated the prototype with raw NL requirements to validate its alignment with the theoretical foundations of the framework and this research study.

The article is structured in different sections. ‘Background and Related Studies’ presents the literature review, and ‘Gamify4LexAmb: Framework for Detecting and Reducing Lexical Ambiguity’ presents the framework in detail. ‘Preliminary Validation of the Proposed Framework Gamify4LexAmb’ discusses the validation of the framework. ‘Discussion’ presents the discussion, and the conclusion of the study is presented in ‘Conclusion’.

Background and Related Studies

In this section, a review of relevant studies on requirements ambiguity and gamification in requirements elicitation is presented.

Ambiguity in natural language requirements

Ambiguity in requirements occurs due to the difference in information articulation between the customer and the analyst (Huzooree & Ramdoo, 2015). This articulation involves words or sentences and becomes more significant when NL-based requirements are specified. Ambiguity is handled by four common approaches, including ambiguity avoidance, detection, reduction, and removal (Osama & Aref, 2018; Sadiq & Jain, 2012).

Beg, Abbas & Joshi (2008) proposed an approach that deals with lexical ambiguity in different phases. In the first phase, it checks the validity of requirements, followed by checking ambiguity in the requirements. Both stages use an algorithm to check the validity and ambiguity. This method applies to the requirements specified in the SRS document. As the domain of NL is quite vast, the proposed approach can be further improved to identify ambiguities efficiently. A semi-automatic method is proposed to recognize ambiguity and inconsistency in SRS (Bajwa, Lee & Bordbar, 2012). The 3-step method has a prototype tool that combines human reasoning and automation for inspections and reviews. The tool parses SRS with constraining grammar and then creates classes, methods, variables, and associations for an object-oriented analysis model. The model is then diagrammed for review and can be used for ambiguity and inconsistency detection. The tool is validated using a case study. The tool works on SRS documents and does not involve users in reducing ambiguity in requirements. Similarly, an approach to minimize ambiguity in NL-based SRS was proposed by Tjong & Berry (2013). The research is based on problems of the informal nature of the English Language. A CNL or controlled natural language representation of requirements is selected with the semantics of business vocabulary and rules (SBVR) to generate accurate and consistent software models. SBVR is machine-processable and creates accurate results. This approach addresses three types of ambiguity: lexical, semantic, and syntactic. However, it does not engage users at the time of elicitation.

In another work, ambiguity is considered a linguistic and cognitive problem (Amna, 2022). A framework is designed to detect ambiguity in user stories to support requirements learning and discovery processes. The work enhances human-related abilities to identify multiple interpretations. Criteria are defined to detect ambiguity in user stories. Currently, there is no implementation and performance evaluation of the framework. Similarly, a tool named TAPHSIR was developed by Ezzini et al. (2022) that detects and resolves anaphoric ambiguity in requirements. The tool focuses on pronouns and aims to identify and revise any pronouns that may confuse development. Machine learning is used for ambiguity detection and a BERT-based approach for anaphora resolution. By analyzing requirements specification, TAPHSIR can determine whether each pronoun occurrence is clear or potentially ambiguous and automatically provide an interpretation for it. The resulting outcome is verifiable by the analysts and requirements engineers. Table 1 presents previous work on ambiguities in NL requirements.

Table 1 Previous approaches of ambiguity detection and reduction in NL requirements.

Ref. & Year	Contribution	RE area	Limitations	Ambiguity Det./Red.	
Ezzini et al. (2022)	Designed theoretical framework
for categorization of ambiguity
in interviews. Based on correct and
incorrect disambiguation.	Requirements gathered
during interviews	No performance
evaluation of the
framework	Detection and reduction	
Popescu et al. (2008)	To enhance the concept of ambiguity with two categories. Focuses on ambiguity in words
and sentences.	Words and sentences	No implementation	None	
Umber & Bajwa (2011)	To show the need for deep
semantics for understanding the
meaning of natural language
requirements with SenseGraph.
It shows information as objects
and events. Helpful in supporting
human activities by automating
RE activities.	Requirements specification	The system is
preliminarily
validated	Detection and reduction	
Vimalraj & Seema (2016)	To identify ambiguity in requirements,
it investigates human cognitive and
analytical abilities with automated
reasoning. The tool pinpoints
ambiguities between different
viewpoints and missing
requirements.	Requirements statements	Experiments conducted
on a small sample
group of students.
No significant difference is
found in the precision	Detection	
Umber & Bajwa (2012)	A theoretical framework for
ambiguity in requirements	Addresses ambiguity in
requirements elicitation
and analysis	Both roles may or may
not recognize ambiguity
present in the requirements	Detection	
Abbasi et al. (2015)	SRAAF helps to write unambiguous
requirements by selecting appropriate
elicitation techniques. Works with W6H
techniques for the evaluation of
different attributes	Selection of elicitation
technique, avoid ambiguities
before writing statements
in an SRS document	The framework was not
fully implemented.
Does not support advanced
technology of NLP for
the application of W6H
techniques.	None	
UNkelos-Shpigel (2018)	To identify ambiguous terms and to
build one linguistic model
for all stakeholders	Requirements elicitation	Showed significant results.
The performance in several
elicitation scenarios
remains unsuccessful	Detection	
Tjong & Berry (2013)	A framework to identify ambiguity in
user stories. Human-centered factors are
identified. The framework is evaluated by
experimenting to test its effectiveness.	Requirements elicitation:
user stories, validation	No implementation	Detection	
Amna (2022)	TAPSHIR is developed for ambiguity
detection and resolution in requirements.
Review pronouns and revise the
pronouns that create misunderstanding	Requirements statements	The practical usefulness of
the tool is unidentified	Detection and resolution	
Fernandes et al. (2012)	Four tools are used with a dataset
of 180 system requirements. Shows
different recall and precision values	NL requirements	Achieving high precision
in pattern detection is
difficult. High recall
is possible, but with
false-negative results	Detection	

Comparison of existing approaches with proposed approach

Table 1 shows that most of the approaches present in the literature address ambiguity detection and do not involve users in reducing or resolving ambiguity. There are a limited number of studies that focus on ambiguity reduction (Popescu et al., 2008; Umber & Bajwa, 2011; Preston, 2014; Vimalraj & Seema, 2016). In previous work, the approaches presented are reactive (address ambiguity once it is detected in the requirements document). Due to a lack of user participation, ambiguity remains unaddressed in the later stages of software development. In some of the studies, both machine learning (ML) and NLP technologies are used to resolve ambiguity, but due to a lack of user involvement in elicitation, communication and collaboration challenges occur between end-users and the development team (Sinha & Husain, 2016; Umber & Bajwa, 2012). Furthermore, platforms built on existing approaches are non-interactive and unresponsive (Abbasi et al., 2015). The proposed approach addresses ambiguity at the stage where requirements are being given and written, i.e., elicitation. It involves users in providing requirements and reducing ambiguities, thus improving the quality of requirements and enhancing collaboration between users and the team.

Gamification in requirements elicitation

Gamification is used in requirements elicitation to involve and engage users in requirements discovery by gamifying various aspects of the activity (UNkelos-Shpigel, 2018). Fernandes et al. (2012) proposed a gamification-based approach, iThink, that uses points/scores to award engaged users upon giving any new requirements. The tool extends stakeholder’s collaboration and participation in the system. It also helps motivate and engage users, but the iThink approach is still considered a first step of gamification in requirements engineering instead of a limited sample size during validation. Another famous work, Requirements Elicitation and Verification Integrated in Social Environment—REVISE (Unkelos-Shpigel & Hadar, 2015), is introduced for collaboration and knowledge sharing among project teams. REVISE works on CARE principles, i.e., create, ask, review, and customer. It uses game elements like scores, leaderboards, and badges to grab user engagement in the activity. Similarly, Lombriser et al. (2016) proposed the GREM—Gamified Requirements Engineering Model, a gamified system that involves users in elicitation for user engagement. GREM uses game elements such as points, badges, levels, leaderboards, feeds, and challenges. The validation of GREM is performed with controlled experiments where user engagement has positively impacted requirements elicitation. Moreover, Kifetew et al. (2017) proposed a DMGame approach based on the Analytic Hierarchy Process (AHP) that uses gamification for requirements prioritization. Game elements such as time, progress, and pontification are used to prioritize collaborative requirements. Table 2 summarizes previous approaches to gamification in RE.

Table 2 Previous studies on gamification in RE.

Ref & Year	Gamification method	Game elements	Limitations	
Dar, Imtiaz & Lali (2023)	A gamified tool for detection of ambiguity from NL requirements, a proactive approach to elicit, verify, and validate user requirements at the same time on the same platform.	Avatar, PBL, ranks, and progress	The tool development, testing, and validation are not mentioned	
Dar, Imtiaz & Lali (2023)	SLR on gamification in elicitation to know which game elements are most suitable for gamified systems for RE, also the challenges of using gamified systems	Points, badges, leaderboards, etc.	SLR identifies only mostly used game elements	
Kolpondinos & Glinz (2019)	GARUSO for the involvement of stakeholders in RE, where stakeholders are not in organizational reach	Points, levels, badges,	Biases, quality of resulting requirements is doubtful, limitations of the approach	
Pimentel et al. (2018)	Ring-i process to perform requirements inspections based on i* models	Rules, cards, goals, feedback	Small sample, inconsistencies in various aspects of the model, acceptance of an idea is unclear, empirical evaluation is required.	
Snijder et al. (2014)	CCRE crowd centric RE method for engaging stakeholders in RE using Refine tool, focus groups were used	Rewards, points, votes,	Not a ‘one size fits all’ solution, negatively influence the trustworthiness of requirements, limited sample	
Alvertis et al. (2016)	CloudTeams Persona Builder is a demo crowdsourcing application based on personas in requirements elicitation.	Levels, badges	No validation does not address user privacy and does not specify how requirements would be specified.	
Lombriser et al. (2016)	GREM gamified the RE model to engage the stakeholders and address the performance. Developed a separate model for requirements elicitation based on user stories	Points, badges, leaderboard, levels, challenges, activity feed	The creativity of the user story was lower, and quality suffered, reducing stakeholders’ communication and collaboration	
Unkelos-Shpigel & Hadar (2015)	REVISE tool based on cognitive theories and implementation of elements of games, designed for the elicitation and verification purposes, based on the principles of CARE i.e., create, ask, review, and extend	Score, leaderboard	A theoretical idea with no implementation, not validated	
Dalpiaz et al. (2016)	CCRE method in SPO was used, and a prototype, REFINE, was built to present the method	Vote, feedback, points	Not well suited for every context, inexperienced, less committed, and untrained team, does not generate desired results, ineffective game elements, another missing element was the ability to merge the needs, participants were behaved and prepared for the use.	
Fernandes et al. (2012)	iThink game-based collaboration tool for improving participation in elicitation	Points	Heavily dependent on the idea generation ability, the test sample was too limited to conclude the study, less visual and less appealing, and had a low rate of amusement, thus generating fewer effective results.	

Gamification provides a solution to user boredom (Hamari & Koivisto, 2015) and uses interactive features to motivate, encourage, and engage (Healey, 2019; Dar, Imtiaz & Lali, 2022) users by using game elements (Mora et al., 2018; Dar, Imtiaz & Ullah Lali, 2022). Some widely used game elements are points, badges, leaderboards, levels, stories, avatars, quests, rewards, ranks, etc. To design gamified systems, three main factors are attached to the games: rules, goals, and feedback system (Gunawardhana & Palaniappan, 2015). Both game elements and game mechanics/rules are important components of any gamification-based system. The game rules in iThink (Fernandes et al., 2012) are designed by awarding 10 points to the users every time a new requirement is provided. Similarly, for new ideas, requirements updates, etc., rules are designed. These game mechanics design strategies on how to use game functions and which game elements will invoke in performing certain tasks. In previous studies, with game elements, game mechanics establish an engaging, interactive, and participatory experience for the users so they can perform the desired tasks without losing any fun. In the next section framework design of our gamified system is given, where we have used points, leaderboard, and levels according to the design and user’s tasks division in the activity.

Gamify4LexAmb: framework for detecting and reducing lexical ambiguity

In literature, limited approaches are proposed that identify and eliminate lexical ambiguities during the requirements engineering phase. Also, the existing approaches identify ambiguities by processing SRS documents, requiring handsome rework and additional cost. Moreover, to date, according to our knowledge, we have yet to identify approaches that focus on recovering ambiguities in the requirements during elicitation by involving stakeholders. Therefore, in this study, we proposed a gamification-based that identifies and resolves ambiguity in requirements at the stage where the users are involved, i.e., elicitation, rather than addressing ambiguity at the inspection phase. We have designed a framework, Gamify4LexAmb, for detecting and reducing ambiguity in NL requirements during elicitation. The proposed framework comprises three major components, including user requirements, gamification, and ambiguity detection and reduction, as shown in Fig. 2.

Figure 2 Gamify4LexAmb framework for detecting and reducing lexical ambiguity.

Created using draw.io.

As shown in Fig. 2, the user (client) provides NL requirements, which are checked for ambiguity. Ambiguity is detected based on two steps: (1) POS tags (Kanakaraddi & Nandyal, 2018) to identify nouns, word dictionaries, and word embedding, and (2) ambiguity rules (Chaudhry & Imtiaz, 2020) to detect lexical ambiguity in NL requirements. The user is given points for providing requirements, and the leaderboard for each user is maintained. For Gamify4LexAmb, game elements are selected by analyzing the literature study and their evaluation with the experts for the effectiveness of the proposed approaches in motivating and engaging users for various software engineering-related tasks using game elements. In the literature, Burke (2012) use points, badges, and leaderboards as game elements to engage the users, and they found better results with stakeholder engagements. Also, Dar, Imtiaz & Lali (2023) use the level game element in addition to the previous three game elements and get encouraging results for user engagement for identifying ambiguities in user requirements. Similarly, Fernandes et al. (2012), Unkelos-Shpigel & Hadar (2015), and Lombriser et al. (2016) used points/scores, leaderboard, PBL, levels and point game elements for their respective REVISE (Unkelos-Shpigel & Hadar, 2015), GREM—Gamified Requirements Engineering Model (Lombriser et al., 2016), and DMGame (Kifetew et al., 2016) approaches for effective user engagement for various software engineering activities. For this purpose, in the proposed approach, we also utilized the points, levels, and leaderboard game elements based on the evidence from the literature aiming to involve users in the process. Where points present a rewarding strategy upon successfully completing a task. Levels keep the curiosity and excitement of users in the activity. A leaderboard is like a scorecard that displays the points, position or ranking of the users. The game elements motivate user involvement in detecting and reducing ambiguity. Gamification helps to reduce ambiguity in given requirements to produce unambiguous requirements. The analyst can also provide requirements and verify each requirement supplied by the users. Moreover, we have designed game rules that use game elements to define the user’s achievement in the system, motivating other users to actively engage in the requirements and possibly remove or minimize ambiguities. Each component of the Gamify4LexAmb framework is elaborated below.

User requirements

For the proposed Gamify4LexAmb approach, NL requirements are provided by the clients and requirements analysts for software applications. NL requirements are given as input to the Gamify4LexAmb system. The system then checks these NL requirements to detect any lexical ambiguity. Clients and analysts are given points for providing requirements. In case of ambiguous requirements, the system prompts the users to update and provide ambiguity-free requirements. The system displays possible suggestions against the lexically ambiguous words if any ambiguity is detected. It prompts the users to provide the requirement again by eliminating the lexical ambiguities.

Gamification

The Gamify4LexAmb approach involves game rules and game elements for detecting and reducing ambiguity in requirements. The game rules are specific to the users involved in the proposed system responsible for providing and analyzing requirements. The user roles specified for the proposed Gamify4LexAmb system include clients, analysts, and project managers (PMs). Upon completion of each task or set of tasks, users are awarded points, and a leaderboard is maintained for each user so they may compare their scores and motivate them to engage in the system. Also, Gamify4LexAmb comprises different levels that unfold various tasks to be completed by the users. Game rules are given in Table 3 below.

Table 3 Game elements and game rules.

Rule no.	Game rules	Game element	
1.	If the client provides a new requirement	Points are given.
Leaderboard is maintained	
2.	If an analyst provides a new requirement	Points are given.
Leaderboard is maintained	
3.	If the analyst verifies each requirement	Points are given.
Leaderboard is maintained	
4.	If the client updates any requirement after verification	Points are given.
Leaderboard is maintained	

As shown in Table 3, we have formulated four game rules covering adding requirements, verifying requirements, and updating requirements. Table 4 shows the game elements reserved for each user role.

Table 4 Game elements and user roles.

No.	User roles	Game elements	
1.	Client	Points, Leaderboard, Levels	
2.	Analyst	
3.	PM	None	

As shown in Table 4, the PM is part of the system but does not play a part in elicitation activity. Similarly, game elements are used to involve users in performing certain tasks and motivating them to participate in the activity. The same game elements are selected for the client and analyst, i.e., points, leaderboard, and levels, as these game elements are commonly used in gamified systems in RE (Gul et al., 2021; Dar, Imtiaz & Lali, 2023).

The Gamify4LexAmb has primarily three user roles. The client and analyst provide requirements, verify requirements, and validate requirements. The PM initiates the process by adding project, users and assigning user roles according to the project. Table 5 shows user roles and the responsibilities they perform.

Table 5 User roles and functionalities.

User role	Functionality	
Client	Provides requirements, Updates requirements, Review requirements document	
Analyst	Provides requirements, Verifies requirements, Review requirements document	
PM	Adds project details, Assigns user roles	

Table 5 shows user roles and their responsibilities in Gamify4LexAmb. Gamify4LexAmb performs several functionalities, including requirements elicitation, requirements verification, requirements updation, and generation of requirements documents. Furthermore, Fig. 3 demonstrates the process with an example.

Figure 3 Example elaborating Gamify4LexAmb mechanics.

Created using draw.io.

As shown in the figure above, after the user provides NL requirements, Gamify4LexAmb checks the requirements for any lexical ambiguity in it. If lexical ambiguity is detected, the user is given suggestions of alternate words and prompted to update and provide the requirement again. In Level 1, the user is given 10 points for providing ambiguity-free requirements. In Level 2, points are given to users for updating any requirement during requirements verification. In Level 3, the requirements document is generated and reviewed by the users. In this way, users feel motivated and get engaged in the system for performing this activity of detecting and reducing ambiguity in requirements.

Ambiguity detection and reduction

We aim to detect and reduce ambiguity by involving and actively engaging users in the requirements elicitation using a gamification-based approach to reduce the hidden costs of the software applications. When the user gives a requirement to the proposed Gamify4LexAmb, the system checks the given requirement for ambiguity. The first step of ambiguity reduction is detection, which is achieved by combining man-made ambiguity rules and POS tags, word dictionaries, and word embeddings. Below, we elaborated on the process in detail.

Ambiguity rules

Man-made ambiguity rules (Chaudhry & Imtiaz, 2020) are incorporated into the gamified system to detect ambiguity in given requirements, as mentioned in Table 6 below.

Table 6 Ambiguity rules.

Rule no.	Lexical ambiguity rule description	
P1.	The same noun or verb should be used throughout the requirement specification.	
Example	Ambiguous: The user shall be able to log in to the system. The employee can generate reports from software.
Un-Ambiguous: The employee shall be able to log in to the system. The employee can generate a report from the system.	
P2.	Instead of using general terms, use domain-specific and specialized terms (or proper nouns)	
Example	Ambiguous: The employee can make a report from the systemUn-Ambiguous: The admin can generate sales report from the accounting system	

POS tags and word dictionary

The proposed Gamify4LexAmb approach utilizes POS tags. POS tags (Kanakaraddi & Nandyal, 2018) are used to identify proper nouns in each requirement that need to be processed by the Gamify4LexAmb. These POS tags, consisting of nouns, verbs, etc., and ambiguity rules (P1 and P2 given in Table 5) identify lexical ambiguity in given NL requirements, as shown in Fig. 2. The word dictionaries WordNet and custom dictionary for synonyms are also employed. The word dictionaries suggest alternative words as a suitable replacement for lexically ambiguous words in a requirement. After the detection of ambiguity, it is removed from the software requirements provided by the users or analysts.

Preliminary Validation of the Proposed Framework Gamify4LexAmb

Another essential part of this study is the validation of the framework. The validation of the framework is performed in two ways, including (1) review by the domain experts and (2) developing a prototype.

Review by domain experts

For this purpose, we created a checklist of important elements in the framework that need to be validated. We reached domain experts associated with the software industry and requested them to validate the framework. For this purpose, firstly, we created a checklist including different components of the framework (Appendix A). Figure 4 further elaborates on the elements of the framework validation checklist.

Figure 4 Review checklist components.

The checklist contains four parts, i.e., questions on user requirements, gamification, requirements ambiguity flow and connectivity of the questions. Along with the checklist, domain experts are provided with the required artefacts, including game rules, ambiguity rules, the objective of the research, expected outcomes, and development details. Domain experts provided their feedback on the framework and showed satisfaction with the theory behind the framework. Experts suggested developing a prototype for better clarity on implementation details. We reached out to several experts in the industry but received only three responses. Due to slow and insufficient response for framework validation, we also developed a prototype to check the working of our designed framework.

Prototype–Gamify4LexAmb

Gamify4LexAmb is a web-based platform (https://crm.southload.com/gamify/login.php) developed in PHP 7.4, Apache Web Server, and MySQL, using different Python libraries. Client, analyst, and PM interact with the web-based interface to give requirements in NL. Gamify4LexAmb then detects lexical ambiguity in NL requirements and prompts the user to give the requirement again (Appendix B). User involvement is made sure in Gamify4LexAmb by employing game elements points, levels, and leaderboard. Let U denote the set of end-users, A represents the set of software analysts, R indicates the set of software requirements, P stand for the set of game points, L denotes the set of levels, and Lb symbolizes the leaderboard, as follows:

• The function P(u) calculates the total points earned by an end-user u.

• The condition C(r) checks whether software requirements submitted by users or analysts are conflict-free.

• The notation (u, r) → p:U × Requirements → Nsignifies that user u providing requirements r, earn points p if the condition Cr is satisfied.

• Here, U × Requirements represents the Cartesian product of the set of end-users and the number of software requirements, and N represents the points earned by each user, adhering to the criteria of ambiguity-free requirements.

• Similarly, (a, r) → p:A × Requirements → Nimplies that analysts a providing requirements r, earn points p if the condition Cr holds.

• A × Requirements is the Cartesian product of the set of software analysts and the number of software requirements, and N represents the points earned by each analyst under the criteria of ambiguity-free requirements.

• For the function updaterequirements(a, r) → p, a software analyst a updating requirements r earns points p.

• For the function verifyrequirements(a, r) → p, a software analyst a verifying requirements r earns points p.

• Finally, to formally express the Leaderboard Lb, assuming P(u) is the total points obtained by user u and P(a) is the total points earned by analyst a:

• Lb (U,A) ={(u,P (u)) ∣u ∈U }∪{(a,P (a)) ∣a ∈A }

• This represents a tuple containing every user and their total earned points, as well as a tuple containing every analyst and their total points earned through writing, updating, and verifying software requirements.

Stages of prototype development

The prototype of Gamify4LexAmb is developed in two stages including ambiguity detection and ambiguity reduction. Table 7 elaborates on these two stages in detail.

Table 7 Two stages of prototype development.

Stages	Description	
Ambiguity detection	1. Get NL requirements	
2. Use dictionaries (Datamuse, Conceptnet, and WordNet)	
3. Word embedding (Glove, Wor2Vec, SciBERT)	
4. Generate a corpus and make pairs of words	
5. Detect all machine term pairs (synonyms) System provides the list of ambiguous requirements	
Ambiguity reduction	6. The system provides the user (client/analyst) with suggestions (all possible synonyms from the dictionary)	
7. The user will select the best words by the chosen words and update the requirement	

Apart from using the dictionaries, we also composed a dataset of words from the requirements for which the prototype is tested.

Checking Gamify4LexAmb for ambiguity detection and reduction

Selection of pilot project

For checking the prototype for ambiguity detection, we considered a small project of the Flour Mill Management Information System (FMIS). The project FMIS was chosen as a pilot project to test the performance of the Gamify4LexAmb tool in identifying ambiguity detection and reduction in natural language requirements. For the proposed approach, the FMIS project is considered as a toy example to demonstrate Gamify4LexAmb working. The selection was made based on the suitability and relevance to our research objectives. FMIS involves user roles (admin, accountant, etc.) and vast functionality (registration, sales management, reports, etc.) that exhibited lexical ambiguity, thus making it a suitable case for validating the Gamify4LexAmb tool. Other selection factors such as well-defined scope, complexity level, variety of functionalities and requirements, and relevancy to the real world also provided a robust test bud for the gamified tool. Admin and accountant are responsible for carrying out various functionalities such as adding suppliers, generating all kinds of forms, managing employees, updating sales information, etc. Gamify4LexAmb prototype is given 29 NL requirements to check if it detects ambiguity according to both rules P1 and P2. Table 8 presents some of the requirements given to the Gamify4LexAmb prototype and ambiguity detection against each requirement.

Table 8 FMIS NL requirements.

#	NL requirements	Ambiguity detection	Lexical ambigutiy free requirements	
		P1	P2	Synonyms		
1.	The user shall log in to the system	✓	X	software system	The user shall log in to the software system	
2.	The accountant shall be able to generate cash payment and cash receipt voucher	✓	✓	None	The accountant shall be able to generate cash payment and cash receipt voucher	
3.	Admin and accountant shall be able to generate bank vouchers	✓	X	registered professional accountant	Admin and registered professional accountant shall be able to generate bank vouchers	
4.	The accountant shall view the cash book and ledger	✓	✓	None	The accountant shall view the cash book and ledger	
5.	Admin and accountant shall be able to generate expense summary reports	✓	X	registered professional accountant	Admin and registered professional accountant shall be able to generate expense summary reports	
6.	Admin shall generate forms for purchases	✓	X	wheat purchases	Admin shall generate wheat purchase forms for wheat purchases	
7.	Admin shall update and add suppliers	✓	X	wheat suppliers	Admin shall update and add wheat suppliers	
8.	Admin shall generate bags issue and return entries	✓	✓	None	Admin shall generate bags issue and return entries	

Table 8 shows eight sample requirements taken from FMIS. Gamify4LexAmb detected five lexically ambiguous words in eight requirements. Overall, it detected 10 ambiguous words in 29 requirements and suggested synonyms for each lexically ambiguous word.

Similarly, clients and analysts are given 10 points for each requirement they provide. Similarly, 10 points are given when the user verifies and updates any requirement. The leaderboard for each user is maintained and updated with the points, as shown in Fig. 5.

Figure 5 Leaderboard displaying Points and Levels.

Created using draw.io.

Discussion

In this study, we have presented a novel approach to detecting and reducing ambiguity in NL requirements. A framework is designed for a gamified system aimed at detecting and reducing ambiguity in requirements during elicitation. The use of gamification is intended to encourage user involvement interactively, to detect and reduce ambiguity in NL requirements collected from end-users. We have also validated the framework from industry experts and developed a prototype. The outcome shows that our approach is helpful in reducing the burden of cost and time constraints in the later stages of software development by addressing ambiguity in the initial stages.

In this study, two research questions are formulated and answered. In RQ-1, a framework Gamify4LexAmb is designed that combines POS tags, word dictionaries and ambiguity rules with gamification techniques to involve users in detecting and reducing ambiguity in NL requirements (Fig. 2). The framework has three major components, including user requirements, ambiguity detection and reduction, and gamification. The client provides requirements for the gamified system. The requirements are checked for lexical ambiguity in the given requirements. The project manager will supervise the whole activity to ensure that the system is implemented effectively and working accordingly. The client is given points for providing requirements. At the same time, the leaderboard of each user is also maintained. Gamification uses game elements and game rules to make elicitation activity more enjoyable for the users. The analyst also provides and verifies requirements. Gamification helps to reduce ambiguity in NL requirements by making elicitation more fun and enjoyable.

In RQ-2, a prototype is developed to validate the Gamify4LexAmb framework. A Gamify4LexAmb prototype is web-based and developed in PHP, and MySQL is used as a database. Different Python libraries are used along with POS tags, word dictionaries, man-made ambiguity rules (P1 and P2), game elements (points, levels, leaderboard), and game rules for the user’s achievements. We also considered a small project of FMIS to evaluate the performance of the prototype. FMIS has two user roles, i.e., admin and accountant. The users give a total of 29 NL requirements. Gamify4LexAmb detected 10 lexically ambiguous words in given requirements. Some of the requirements and ambiguous words are presented in Table 8. The prototype not only detected ambiguities but also involved users in the activity by giving points against each requirement. Upon providing requirements, 10 points are given, while 10 points for updating and 10 points for verifying requirements are given to the users. A leaderboard is maintained for each user after performing the desired activity in the system. However, the Gamify4LexAmb prototype is well-aligned with the theoretical foundations of the study.

Threats to validity

This study proposes a novel idea to address natural language requirements with a gamification-based approach. However, the study has faced threats to internal and external validity, which are discussed below.

Threats to Internal Validity:

• The selection of game elements is an internal threat to validity. Although we have presented evidence from literature that points, levels, and leaderboards are significant for user engagement in elicitation, it still needs to be validated by industry experts. Also, in the future, other game elements will be included to utilize the benefits of gamification in elicitation.

• Another threat was the selection of a pilot case/project for testing Gamify4LexAmb. We tried to minimize the threat’s impact by aligning it with the research objectives.

• Currently, only a limited number of users can access the system, but the gamified system is multiplayer. In the future, we aim to validate the proposed approach with larger software development groups, comprising requirements for larger software projects. Additionally, we are planning to validate the proposed approach to crowd-based requirements engineering (Khan et al., 2022).

• External Validity:

Due to the limited number of domain experts for validation, the prototype based on preliminary findings, and the new area of gamification, it is hard to draw broader conclusions at this stage. Therefore, as mentioned earlier, in the future, the approach will be validated on crowd-based RE and larger software development groups.

Conclusion

Gamification is a unique way of involving users in detecting and reducing ambiguity in NL requirements. In this study, a framework Gamify4LexAmb is designed, and the prototype is developed to detect and reduce ambiguity in NL requirements. The prototype incorporates three game elements—points, leaderboard, and levels—that are assigned to involve two different user roles: client and analyst. The framework is based on lexical ambiguity in requirements. Ambiguity rules have been identified and incorporated into the system design, along with game elements and game rules. Gamify4LexAmb is validated by domain experts from the software industry. For this purpose, a checklist is designed comprised of four parts, i.e., user requirements, gamification, ambiguity detection and reduction, and flow and connectivity of all design components. The domain experts suggested developing a prototype to evaluate the performance of the framework. A prototype is also developed to check whether lexical ambiguity is detected and reduced by the gamified system. Gamify4LexAmb is checked on raw requirements from a small project. The outcome shows that it can detect ambiguity in NL requirements and suggest users for possible word alternatives. Not only this, but Gamify4LexAmb also involves and engages users by employing fun-based game elements.

The future work involves the selection of an appropriate case study from an IT company, suitable projects, and stakeholders to be involved in the gamified system. Currently we are improving the accuracy of the tool to efficiently address ambiguity in given requirement statements. Also, improvements to the user interface are being made. We acknowledge the strength of using large datasets for validation purposes. There, we aim to customize the Gamify4LexAmb approach for Marked-based software applications, where requirements for the software products are gathered from a large pool of various social media users (Khan et al., 2019a). In these platforms, users use free language to propose possible requirements cum features (Khan et al., 2019b). There will be a high chance of ambiguous requirements, and the proposed customized Gamify4LexAmb tool will be useful in identifying ambiguities and engaging end-users to actively participate in requirements. Furthermore, various natural language processing and machine learning approaches will be adopted to extract useful information from the end-user feedback before feeding into the customized Gamify4LexAmb tool (Hassan et al., 2024). Additionally, we plan to conduct a case study from the software industry to gain the benefits of using gamified tools for reducing ambiguities by employing Gamify4LexAmb in a real-time software case study. Once developed, this gamified system has the potential to revolutionize requirements elicitation in software development projects. The use of gamification in requirements elicitation helps to make elicitation more engaging and interactive, which in turn leads to more accurate and complete requirements.

Supplemental Information

Supplemental Information 1 Appendix

(A) Review Checklist, (B) P1 and P2

Supplemental Information 2 Code Prototype

Supplemental Information 3 Raw data

Supplemental Information 4 Validation Checklists

Additional Information and Declarations

Competing Interests

Author Contributions

Data Availability

The authors declare there are no competing interests.

Hafsa Dar conceived and designed the experiments, performed the experiments, analyzed the data, performed the computation work, prepared figures and/or tables, and approved the final draft.

Romana Aziz conceived and designed the experiments, performed the experiments, analyzed the data, authored or reviewed drafts of the article, and approved the final draft.

Javed Ali Khan conceived and designed the experiments, performed the computation work, prepared figures and/or tables, authored or reviewed drafts of the article, and approved the final draft.

Muhammad IkramUllah Lali conceived and designed the experiments, performed the computation work, authored or reviewed drafts of the article, and approved the final draft.

Nouf Abdullah Almujally performed the experiments, analyzed the data, authored or reviewed drafts of the article, and approved the final draft.

The following information was supplied regarding data availability:

The prototype link and review checklists are available in the Supplementary Files.

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
