# Peer review of "Gamify4LexAmb: a gamification-based approach to address lexical ambiguity in natural language requirements"

_PeerJ Computer Science, doi:10.7717/peerj-cs.2229_

## Round 0.1 · original submission · Major Revisions

Both reviews highlight the need for more explanation of the tool's design and validation. The paper should justify its choices for a specific ambiguity taxonomy and game elements. Including a comparison with existing methods and showcasing the tool's impact on requirement clarity would strengthen the research. Finally, discussing future plans to validate the tool's effectiveness in real-world scenarios would improve the conclusion.

Please see the reviewers' comments.

Reviewer 1 ·

Basic reporting

The authors developed a prototype of gamification-based tool for addressing lexical ambiguity in NL requirements. Gamify4LexAmb incorporates lexical ambiguity rules, Parts of Speech (POS) tags, game rules, and game elements (points, leaderboard, etc.) to elicit requirements better by engaging users. The topic they have covered is interesting and very significant for RE community.

Experimental design

Regarding Figure 1 showing types of requirements ambiguity, why authors have selected this taxonomy despite having other taxonomies? There is no clear justification for selecting mentioned taxonomy.

In section 1.C, why only lexical ambiguity in NL requirements is addressed? It would have been better to give reason for selecting lexical ambiguity.

In section 2, the table showing previous approaches on ambiguity detection and reduction I missed comparison of existing works with the authors' proposed tool. Highlighting the difference with previous work could be added.
What is the reason for adopting only three game elements and how these elements were extracted in the methodology of the proposed tool? There is no justification mentioned in the paper.
In section 4.D a small project FMIS is used for checking ambiguity detection and reduction features but there are no details given on how this project is selected? Are the requirements published?

Validity of the findings

In the conclusion, the authors did not share the progress in future work. It is important to add, since in my understanding, monitoring the use of the tool, once developed, and validated, can verify (such as its effectiveness in addressing ambiguity in the requirements elicitation)

Reviewer 2 ·

Basic reporting

This paper is based on an interesting concept and presents an approach to address ambiguities in Natural Language Requirements. The section 1 and 2 clearly states the purpose of the study.

Experimental design

Gamification in any system includes inclusion of game elements and game mechanics. In background authors have covered game elements used in RE but game mechanics or game rules are not covered in detail. The coverage of game e mechanics/rules in previous studies must be covered in the background.

Can you please tell me how did you decided on the game elements? What is the reason of selecting only points, levels, and leaderboard? I didn’t find any justification in the paper.

Validity of the findings

The tool was preliminary validated by 3 experts from the industry which I think are very few to validate a tool specifically requirements ambiguity. How this proposed approach Gamify4LexAmb is adequate for addressing ambiguity, detection, and reduction precisely for any software?

It would be nice to show the requirements after ambiguity resolution to see the difference in Table 8

The authors have used two words consequently ‘addressing’ and ‘reduction’ ambiguities. The paper title refers towards addressing the ambiguity but the body of the paper completely states ambiguity detection and reduction. This leads to confusion.

---

## Round 0.2 · accepted · Accept

I confirm that the authors have addressed all of the reviewers' comments. One previous reviewer declined the invitation, so I have assessed the revision myself.

Reviewer 1 ·

Basic reporting

The authors incorporated most of my previous comments, however, below is a minor suggestion.


To increase the paper's readability, it is recommended to convert the abstract to a structured format, i.e., background, problems, methodology, results, and conclusion

Experimental design

The authors have incorporated the comments suggested to them.

Validity of the findings

The authors have incorporated the comments suggested to them.